# High-Pressure Treatment in Combination with Reduced Sodium for Improving the Physicochemical Properties and Sensory Qualities of Pork Gels

**DOI:** 10.3390/foods14010096

**Published:** 2025-01-02

**Authors:** Weitong Wang, Jingying Cai, Satomi Tsutsuura, Tadayuki Nishiumi

**Affiliations:** Graduate School of Science and Technology, Niigata University, 8050 Ikarashi 2 nocho, Nishi-ku, Niigata 950-2181, Japan; f22m006c@mail.cc.niigata-u.ac.jp (W.W.);

**Keywords:** high hydrostatic pressure, low-sodium meat product, physicochemical property, sensory evaluation

## Abstract

High-pressure treatment was utilized in this study to produce high-quality, reduced-sodium pork gels with desirable texture and sensory properties, addressing the challenge of maintaining quality in low-sodium meat products to meet health-conscious consumer demands. High-pressure treatment applied within the range of 150–200 MPa significantly reduced cooking loss while maintaining moisture content and provided an ideal network structure for reduced-sodium pork gels. High-pressure treatment at up to 100–200 MPa, in combination with added sodium chloride and sodium polyphosphate, was evaluated for its effects on gel texture, with results indicating that high-pressure treatment significantly improved breaking stress (increased by 10.01% under 150 MPa and 14.66% under 200 MPa), modulus of elasticity (increased by 14.77% under 150 MPa and 24.17% under 200 MPa), and hardness (increased by 11.12% under 150 MPa and 11.45% under 200 MPa). Rheological characteristic measurements revealed that gel strength was highest at 150 MPa (G′ = 443,000 Pa; G″ = 66,300 Pa and tanδ = 0.15), which showed higher G′ and G″ values and similar tanδ compared to the 0.1 MPa, 2% NaCl + 0.5% SPP condition (G′ = 334,000 Pa; G″ = 49,200 Pa; tanδ = 0.148). Protein analysis by sodium dodecyl sulfate–polyacrylamide gel electrophoresis showed a reduction in the α-actinin band with increased pressure, which suggested protein interactions were enhanced. Differential scanning calorimetry analysis indicated that protein denaturation occurred more readily at higher pressures (0.071 J/g at 0.1 MPa, 0.057 J/g at 150 MPa, and 0.039 J/g at 200 MPa). These findings underscore the value of treatment under high pressure at 150 MPa developing reduced-sodium meat products with desirable texture and flavor characteristics.

## 1. Introduction

Meat is a vital food group for most consumers, particularly in developing countries, and provides high-quality protein, essential micronutrients, and various fats, including crucial omega-3 polyunsaturated fats. Factors such as economic status, live-stock production levels, and the socioeconomic conditions of consumers contribute to meat consumption patterns [1,2]. Additionally, demographic factors such as gender, age, religious beliefs, weight-to-height ratio, and total caloric intake influence meat consumption [3]. A high-meat diet has been linked to an elevated probability of developing chronic health issues, including cardiovascular disorders, renal disorders, various forms of cancer, and diabetes mellitus, leading to a growing awareness of the potential health risks linked to excessive meat consumption [4,5,6]. Excess sodium chloride (NaCl) intake is known to increase blood pressure, which is a significant risk factor for cardiovascular and kidney diseases [7]. Between 1982 and 2015, urine samples indicated that the average sodium chloride (NaCl) intake was 9.3 g/day [8]. In contrast, the World Health Organization recommends a maximum intake of 5 g/day [9]. Processed foods account for approximately 77% of sodium consumption [10], with 20–30% originating from meat products [11]. Processed meat refers to products containing at least 30% meat that includes items like manufactured, cured, or dried meats [12]. The characteristic qualities of these meat products rely heavily on the manipulation of protein functional properties during processing, especially through the development of a protein gel matrix. The matrix develops as myofibrillar proteins undergo unfolding, aggregation, and cross-linking during thermal processing, resulting in a continuous solid network that is crucial to the texture and structure of processed meats [13]. Sodium chloride is typically used at 2–6% in meat products [14] and has a sodium amount-of-substance fraction of 39.3% [15]. To more closely simulate market-available meat products, we added sodium phosphate alongside sodium chloride. Both additives are essential for myofibrillar protein extraction and solubilization, which facilitate stable meat gel formation.

In the food industry, 25–30% of all high-pressure-processed foods are meat products [16]. Research indicates that high pressure affects myofibrillar proteins similarly to sodium or phosphate, and this effect of high pressure allows for reduction of the amounts of these additives [17]. High-pressure treatment induces three primary changes in meat: protein modifications (especially in myofibrillar proteins), enzymatic alterations, and structural changes [18]. Treatment at moderately high pressure (approximately 200 MPa) enhances the gelation properties of myofibrillar proteins by modifying their structures and gel microstructures [19]. High-pressure treatment at 100–150 MPa can denature or depolymerize actin, resulting in an increased presence of F-actin within actomyosin; this process helps counteract myosin’s adverse impact on the gel’s structural strength. Applying a pressure of 150 MPa improves the heat-induced gel strength of actomyosin, which directly affects muscle tissue [20]. Furthermore, it has been shown that changes in the high-pressure conditions and cooking methods affect the final texture of meat products [21]. Treatment at moderately high pressure (≤200 MPa) can mildly denature myofibrillar proteins and lead to dissociation of protein quaternary structures and depolymerization of actin and actomyosin [18], which promotes the solubilization of myofibrillar proteins. In contrast, higher pressure treatment (≥300 MPa) can denature myofibrillar proteins and destabilize them. The exposure of hydrophobic and sulfhydryl groups to the aqueous environment facilitates the formation of disulfide bonds and strengthens hydrophobic interactions, which leads to the formation of insoluble aggregates and decreased myofibrillar protein solubility [22]. Vaudagna et al. [23] found that high-pressure treatment at 350 MPa for 6 min at 20 °C hardened pork meat batters without added sodium or phosphate. However, adding sodium chloride (1.5% or 3.0% NaCl) or sodium polyphosphate (SPP) (0.25% or 0.5% ${\mathrm{Na}}_{4}P_{2}O_{7}\cdot 10H_{2}O)$ neutralized this high-pressure effect and gave a softer product texture. Applying high-pressure processing improves the functional characteristics of myofibrillar proteins while reducing the NaCl and SPP mass fraction, and this provides interesting opportunities for processing gel-type meat products.

While prior research has explored the effects of high-pressure processing on meat, studies on lower pressures in reduced-sodium pork gels are limited. To address this research gap, this study aimed to investigate the effects of high-pressure treatment at <200 MPa combined with reduced concentrations of sodium chloride and SPP on the physical and chemical properties of low-sodium, low-phosphate pork gels. The results obtained for commercially available pork gel products could be used to meet the market demand for healthier, low-sodium alternatives without compromising product quality.

## 2. Materials and Methods

### 2.1. Raw Materials

Chilled normal pork leg was sourced from Marudai Food Co. (Osaka, Japan). The muscles used in this study were derived from the Biceps Femoris Muscle located in the inside leg muscle, which was processed by the supplier into 1 kg packages and delivered to the laboratory. Upon receipt, the samples were immediately stored at −20 °C to ensure their quality and integrity. Before experimentation, the meat was left to partially defrost overnight at a temperature of 4 °C, and visible connective tissue and fat were carefully removed using a knife. The pork was then cut into uniform pieces, vacuum-sealed in polyethylene bags (75 μm thickness, 180 × 260 mm, ASAI KASEI ST2540a N8, Tokyo, Japan) using a degassing sealer (vacuum level of −95 kPa, FCB-200, Fuji Impulse, Osaka, Japan), stored at −80 °C for six hours to freeze, and subsequently returned to storage at −20 °C. All experiments were conducted using this frozen pork. For each trial, a portion of the pork was left to thaw at 4 °C overnight and subsequently utilized for pork gel preparation. The muscle pH, measured with a pH meter (ø 11 mm, length: 35 mm, plug-in probe, TESTO 206-pH2, Yokohama, Japan), ranged from 5.55 to 5.70.

### 2.2. Preparation of Pork Gels

Initially, 1.0 kg of deboned pork was ground using a meat grinder (BK-220 kitchen mincer, Bonny, Tokyo, Japan) equipped with 16.0 mm and 3.2 mm grinding discs. The ground meat was then combined with specific concentrations of sodium chloride (0–2% NaCl) and/or SPP (0–0.5% ${\mathrm{Na}}_{4}P_{2}O_{7}\cdot 10H_{2}O)$ in a food processor (National MKK78-W, Panasonic, Osaka, Japan) for approximately 60 s. Subsequently, 20 g of the mixture was transferred into three metal molds (ø 40 mm, height: 15 mm) to form pork gels under the conditions specified in Table 1. To ensure there was a sufficient sample volume for subsequent analyses (color, texture, and sensory evaluations), each experiment was conducted in triplicate. Treatments were conducted using samples from three different pork legs, and triplicate samples were prepared for each pork leg. The temperature of the final mixture was maintained at 4 °C until required for further processing. The samples were individually vacuum-sealed in polyethylene bags (75 μm thickness, 180 × 260 mm, ASAI KASEI ST2540a N8, Tokyo, Japan) using a degassing sealer (vacuum level of −95 kPa, FCB-200, Fuji Impulse, Osaka, Japan). Each vacuum-sealed bag was then placed in a larger air-free polyethylene bag (75 μm thickness, 250 × 400 mm, ASAI KASEI ST2540a N9, Japan) that was filled with water and sealed. The pork gels were subjected to high-pressure hydrostatic processing for 10 min at 20 ± 5 °C in a high-pressure food processor (Dr. CHEF, Kobe Steel Ltd., Kobe, Japan) at 100, 150, and 200 MPa. Control samples were treated at atmospheric pressure (0.1 MPa). For technical characterization, samples were submerged in a water bath and cooked at 80 °C for a duration of 30 min. This was followed by cooling in ice water until the core temperature reached 20 °C (approximately 15 min).

### 2.3. Meat Color

The meat color was assessed using the 1976 CIELAB system, and the L* (lightness), a* (redness), and b* (yellowness) values were measured (aperture opening size: 8 mm, optical geometry: diffuse illumination/8° viewing, observer angle: 2° standard observer, Illuminant type: CIE Standard Illuminant D65, CR-400, Konica Minolta, Tokyo, Japan). Before each measurement series, the chromameter was calibrated with a white ceramic standard. Color measurements were conducted at 4 °C under standard atmospheric conditions on both raw and thermal gels across 24 treatment groups (6 formulations × 4 high-pressure treatments), with three replicates per group.

### 2.4. Water Content and Cooking Loss

The water content of the samples was measured using a halogen moisture analyzer (HG63, Mettler Toledo, Zürich, Switzerland). Around 3 g of each heated pork gel was subjected to heating at 137 °C for 90–110 min. Measurements were taken from 24 groups for each of the raw and heated pork gels, with three replicates per group.

The cooking loss of pork gels was calculated based on the weight difference recorded before and after heating, using a standard calculation (Equation (Equation 1)). This evaluation was conducted across 24 treated groups of heated pork gels, with three replicates per group.
(1)$$Cooking\;loss\left(\%\right)=\frac{\left(Mass(raw)-Mass(heated)\right)}{\left(Mass(raw)\right)}\times 100.$$

### 2.5. Breaking Stress and Modulus of Elasticity

Measurement of breaking stress and modulus of elasticity of the heated pork gels using a creepmeter (Creep Meter RE2-33005B, Yamaden, Tokyo, Japan) at ambient room temperature. The pork gels, cut to 2.0 × 2.0 × 1.0 cm, were used to puncture with a spherical plunger (ø 5 mm) at a constant speed of 1 mm/s with a 100% compression ratio. The breaking stress, which is the force required to fracture the meat sample, was used as an indicator of gel strength. The elastic modulus, which is defined as the slope of the stress–strain curve just before fracture, was used as an indicator of the elasticity of the pork gel. Measurements were conducted for 24 treatment groups of thermal pork gels, with three replicates analyzed for each group.

### 2.6. Texture Profile Analysis

Texture profile analysis (TPA) of the heated pork gels was performed with the help of a creepmeter (Creep Meter RE2-33005B, Yamaden, Tokyo, Japan). The pork gels were diced into cubes measuring 1.5 × 1.5 × 1.0 cm and subjected to compression with a disk plunger (ø 40 mm) at a speed of 10 mm/s to apply a 60% compression ratio through double compression. Key parameters of hardness, cohesiveness, and adhesiveness were evaluated for each gel sample. The hardness (unit: Newton) represents the peak force during the initial compression cycle. Cohesiveness, which reflects the strength of internal bonds in the pork gels, was calculated as the ratio of the positive force area during the second compression cycle to that of the first. The adhesiveness (measured in Newtons) was determined by analyzing the negative area during the first compression cycle. This area corresponds to the energy needed to overcome the attractive forces between the probe and the gel, serving as an indicator of the gel’s viscosity. TPA was performed on 24 treatment groups of thermal pork gels, with three replicates analyzed for each group.

### 2.7. Rheological Characteristics

The gelation characteristics of the meat samples were assessed using a rheometer (Physica MCR301, Anton Paar, Graz, Australia) with a parallel plate geometry (PP25/P2, ø 24.987 mm). Measurements were conducted at a strain of 1% and constant frequency of 10 Hz, and with a 3 mm gap between the plates. The temperature was gradually increased from 20 °C to 90 °C at 1 °C/min. Throughout the process, both the storage modulus (G′) and loss modulus (G″) were continuously recorded, and the ratio of the loss modulus to the storage modulus (tanδ) was computed.

### 2.8. Observation by Scanning Electron Microscopy

The pork gels were initially pre-fixed using a glutaraldehyde solution, followed by treatment with a tannic acid solution. This procedure was based on the osmium method. The samples were then examined with a scanning electron microscope (JSM6510LA, Jeol Ltd., Tokyo, Japan) operating at an acceleration voltage of 15 kV and using the SS40 mode for observation.

### 2.9. Differential Scanning Calorimetry (DSC)

The thermal behavior of proteins in the pork gel was analyzed using a micro differential scanning calorimeter (Micro DSC VII, Setaram, Caluire-et-Cuire, France). Approximately 300 mg of each sample was sealed in a stainless-steel container and heated from 4 °C to 90 °C at a rate of 1 °C/min. An equal mass of water was used as a reference. The transition temperature (Tm) was recorded, and the enthalpy of transition (ΔH) for each peak was calculated from the peak area using Setaram software, Version ASTM A698 and expressed in joules per gram. Additionally, the total enthalpy of transition was determined from the peak area, while the total enthalpy of change (ΔHT) was derived by constructing a baseline between 45 °C and 80 °C and calculating from the corresponding peak areas.

### 2.10. Muscle Protein Solubility

The muscle protein contents in both dissolved and undissolved protein components were measured using an established method [19]. For this purpose, 1 g of the pork gel was combined with 20 mL of an equiconcentrated (20 times the sample volume) NaCl solution and homogenized for about 60 s using a microhomogenizer (NS-310 E11, Microtec Co., Ltd., Chiba, Japan). Subsequently, the mixture was centrifuged at 20,000× *g* for 30 min at 4 °C. The supernatant obtained by filtration was determined for protein concentration using the biuret method. The composition of the biuret reagent is shown in Table 2. For the insoluble fraction, 0.1 g of the precipitate was dissolved by mixing with 2 mL of 10% NaOH (20 times the sample amount), and the protein content in the precipitate was also determined by The biuret assay. The solubility was derived using Equation (Equation 2). Three replicates were analyzed for each group.
(2)$$Muscle\;protein\;solubility\left(\%\right)=\frac{\left(Soluble\;protein(mg/g)\right)}{\left(Soluble\;protein+Insoluble\;protein(mg/g)\right)}\times 100$$

### 2.11. SDS-PAGE Analysis

Meat samples were prepared following an established method [19]. Initially, 1 g of pork gel was combined with 20 mL of an equiconcentrated (20 times the sample volume) NaCl solution and blended for approximately 60 s using a microhomogenizer (NS-310 E11, Microtec Co., Ltd., Chiba, Japan). The protein composition of this fraction was analyzed by sodium dodecyl sulfate polyacrylamide gel electrophoresis (SDS-PAGE) following an established method [24]. Gel electrophoresis was carried out using a 10% polyacrylamide gel concentration with a 10% separating gel using an appropriate electrophoresis buffer. Two different electrophoresis patterns were generated. One pattern had a uniform application of 2.0 µL per well, and in the other, the protein was adjusted to 10 µg per well.

### 2.12. Statistical Analysis

For the analysis of color, cooking losses, and TPA, statistical comparisons were conducted using SPSS software (SPSS Inc., Chicago, IL, USA, Ver 22.0). Statistical significance between results was tested through multifactorial analysis of variance, followed by Tukey’s multiple range test. Statistical analyses, including variance analysis, mean calculations, and standard error determinations, were conducted using Microsoft Excel 2010 (Microsoft Office Excel 2010 for Windows). Differences between treatments were considered statistically significant at *p* < 0.05.

## 3. Results and Discussion

### 3.1. Meat Color

A meta-analysis by Ha et al. [25] on the effects of high-pressure treatment (50–215 MPa) on fresh meat color revealed that this process greatly increased brightness (L*) and reduced redness (a*). This color change is thought to occur through two primary mechanisms: (1) protein coagulation, which reduces sarcoplasmic and myofibrillar proteolysis and alters meat matrix, and (2) degradation of myoglobin through the displacement or liberation of the iron porphyrin complex. The decrease in red color after high-pressure treatment results from the oxidation of ferrous myoglobin to brown ferric methemoglobin and myoglobin denaturation [25]. These results are consistent with those from Maksimenko et al. [26] on beef gel appearance. High-pressure treatment (100–200 MPa) applied to pork gels without sodium chloride or SPP produced L* values similar to the untreated control (0.1 MPa), with slight increases in the a* and b* values (Table 3). For samples containing NaCl and SPP, high-pressure treatment contributed to moderate increases in the a* and b* values compared with the control. Samples with low NaCl and SPP concentrations that were subjected to high-pressure treatment showed slight increases in their L* values and decreases in a* values relative to the control containing 2% NaCl + 0.5% SPP (0.1 MPa) (*p* < 0.05). Zhu et al. [27] found that reducing the sodium mass fraction in sausages greatly increased the L* value and decreased the a* value, while high-pressure treatment further increased the L* value. This increase in brightness and whiteness was attributed to myofibrillar and sarcoplasmic protein denaturation and precipitation during high-pressure treatment, and increased oxidation resulting in heme replacement and methemoglobin denaturation [28].

### 3.2. Measurement of Water Content and Cooking Losses

A significant effect of high-pressure processing on muscle proteins is its alteration of the actomyosin interaction, with structural reorganization or interruption within the myofibrillar fibers influencing the water retention capacity [17,19]. Table 4 shows the effect of high-pressure treatment on the water content and cooking loss of pork gel. Compared with the control (0.1 MPa), high-pressure treatment had no significant effect on either the water content or the cooking loss in thermal pork gels. This aligns with the observation of Iwasaki et al. [19] that sodium addition controlled pressure-induced cooking loss. In this research, the addition of NaCl and SPP resulted in a modest increase in water content and a notable decrease in cooking loss (*p* < 0.05) in gels treated at 150–200 MPa compared with the untreated samples. These findings are consistent with those of Maksimenko et al. [26], who noted that cooking loss in beef gels significantly decreased (*p* < 0.01) with the addition of NaCl and/or SPP. Without sodium chloride, high-pressure treatment alone did not improve water retention in pork gels; however, in combination with NaCl and SPP, high-pressure treatment significantly enhanced water retention and reduced cooking losses. Notably, cooking losses in control gels containing 2% NaCl + 0.5% SPP were similar to those in gels treated at 150 MPa with 1% NaCl + 0.5% SPP. Pressures above 150 MPa induced substantial structural changes in myogenic fibers, which likely contributed to reductions in the water-holding capacity at pressures exceeding 150 MPa [29].

### 3.3. TPA

The effects of high-pressure treatment and sodium chloride and SPP addition on the breaking stress, elastic modulus, hardness, and cohesiveness of the thermal pork gels are shown in Table 5. Compared with control pork gels (0.1 MPa), high-pressure treatment at 100–200 MPa resulted in a significant increase in the breaking stress of the pork gel (*p* < 0.05). Regardless of NaCl and SPP concentrations, their presence in unpressurized samples reduced the breaking stress, elastic modulus, and hardness, while these properties were enhanced under high-pressure treatment. Sikes et al. [30] reported that low-sodium (1%) beef sausages exhibited improved texture and sensory acceptability after high-pressure treatment (up to 400 MPa for 2 min at 10 °C) and cooking (76 °C for 25 min), with water retention optimized at 200 MPa. This increase was attributed to the formation of a denser, more uniform network structure in myofibrillar proteins, likely resulting from HPP (≤200 MPa) dissociating or depolymerizing filament structures, which disrupted and dispersed them into shorter filaments, thereby facilitating the development of a stronger gel network [30]. Moreover, cohesiveness was also significantly higher (*p* < 0.05) in samples treated at 150–200 MPa with 1% NaCl than in those treated with 2% NaCl + 0.5% SPP. The bond strength between meat particles is a critical factor in meat-product manufacturing, and high-pressure treatment can improve the cohesiveness of meat particles in comminuted meat products, particularly at reduced sodium concentrations [31]. Control pork gels containing 2% NaCl + 0.5% SPP treated at 0.1 MPa exhibited lower gel strength than those containing 1% NaCl + 0.5% SPP and treated at 100–200 MPa (*p* < 0.05). Zhang et al. [22] found that high-pressure treatment (100–200 MPa) disrupted the protein structure stability, which promoted stronger protein–protein interactions and contributed to the formation of a uniform gel microstructure and increased gel hardness.

### 3.4. Rheological Characteristics

Our experiments on pork gels showed that high-pressure treatment at 150–200 MPa before heating is effective in improving water retention and reducing cooking loss. Additionally, for reduced-sodium pork gels containing 1% sodium chloride, high-pressure treatment had positive effects on the breaking stress, elasticity, hardness, and cohesiveness. Therefore, subsequent experiments were simplified by halving the sodium mass fraction (1% NaCl) compared with commercial products (2% NaCl) and applying high-pressure treatment at 150–200 MPa to evaluate the gelation and sensory characteristics of the pork gels.

The viscoelastic properties of myosin were assessed by measuring the storage modulus (G′, Figure 1a), loss modulus (G″, Figure 1b), and phase angle (tanδ = G″/G′, Figure 1c). The G′ value represents the stored energy in the elastic component and reflects the gel strength [32], while the tanδ value describes viscoelastic behavior and reflects the transition from a viscous myosin solution to an elastic myosin gel [33].

As the temperature increased to approximately 50–58 °C, the G′ and G″ values of the pork gel gradually rose, which suggested gel strengthening. At this stage, most of the proteins had unfolded, with increased cross-linking forming a network that strengthened the gel matrix, and this resulted in a robust and irreversible gel structure [34,35]. At 150 MPa, the G′ and G″ values of sodium-free and reduced-sodium pork gels exceeded those of untreated samples, which indicated that moderately high-pressure treatment promoted the formation of more elastic gels. This was likely because of the increased exposure of R-SH and hydrophobic groups in the high-pressure-treated myosin, which would enhance cross-linking during heating [36]. However, a further increase in the pressure to 200 MPa reduced both the G′ and G″ values and increased tanδ. This change indicated that excessive pressure adversely affected the gelling capacity of myosin. Similar trends were observed by Chen et al. [37] in myosin–κ–carrageenan gels treated at high pressure (0–400 MPa), where moderate pressures (≤150 MPa) induced mild protein unfolding but most myosin molecules retained their native structure. Under such conditions, the protein unfolding rate exceeded aggregation as the temperature rose, which promoted a compact, uniform gel structure. Changes in G′ and G″ are closely tied to protein denaturation and aggregation levels [32].

There was no notable variation in G′, G″, or tanδ values across the 0–200 MPa range, which suggested that high-pressure treatment at <200 MPa did not significantly alter the viscoelastic characteristics of the continuous gel network. At higher pressures, complete unfolding and aggregation inhibited further unfolding during heating and this reduced elasticity. Cao et al. [32] attributed the decline in G′ to increased surface hydrophobicity, which compensated for reduced G′ after myosin denaturation under high-pressure treatment. The texture properties of protein gels are influenced by their microstructures, which are largely determined by the balance between protein unfolding and aggregation. Rapid aggregation relative to unfolding creates a dense, uniform gel, while slower aggregation yields a course, heterogeneous structure [38]. Under moderate pressures (≤200 MPa), proteins undergo mild unfolding, with most molecules retaining their native configurations. During subsequent thermal gelation, the unfolding rate surpasses the aggregation rate, and this promotes a uniform gel microstructure with enhanced hardness. In thermal gelation, protein denaturation and unfolding generally increase G′ and G″; however, as high-pressure treatment increases, more protein denatures, which reduces the number of native molecules available to strengthen the gel upon heating and decreases G′ and G″. Therefore, protein denaturation induced by high-pressure treatment directly contributes to reductions in G′ and G″ [22]. Overall, our results showed that treatment at 150 MPa and 200 MPa enhanced gel strength through different structural changes at varying levels.

### 3.5. Observation by Scanning Electron Microscope

Figure 2 shows the microstructures of the thermal pork gels. In the untreated pork gels without sodium chloride and/or SPP addition (sample S1), many fragmented muscle fibers and coarse aggregates were evident. This suggests that myofibrillar proteins coagulated upon heating and formed large clumps. This result is consistent with the findings of Tintchev et al. [39], who observed a rougher matrix in reduced-NaCl frankfurter batters compared to conventional batters, indicating a less uniform structure. Typically, solubilized myofibrillar proteins, primarily myosin, form a three-dimensional network structure after heating [40]. Compared with S1, the microstructures of pork gels with added sodium chloride and SPP (samples S2–S5) exhibited higher homogeneity and fewer interstitial spaces, which resulted in a denser and more uniformly distributed three-dimensional mesh. Water and fat will be retained within this network, whereas a denser network holds a greater amount of water and fat and contributes to enhancing the water retention ability and structural integrity of the meat gels [41]. With high-pressure treatment within the range of 150–200 MPa (samples S2 and S3), the microstructure transitioned further and formed an improved dense and homogeneous network. In the samples with low sodium and treated at high pressure (samples S3 and S4), a distinct transformation from a fibrous structure to a three-dimensional lattice with larger pores was observed compared with the high-sodium S5 samples. The application of high pressure (150–200 MPa) appeared to influence protein unfolding, contributed to a reduction in myofibrillar protein particle size, and enabled a uniform alignment during subsequent heating. These changes promoted a compact and consistent gel network structure. Overall, the high-pressure treatment within the range of 150–200 MPa significantly improved the microstructural uniformity and density of the pork gels, especially in samples with reduced sodium.

### 3.6. Differential Scanning Calorimetry (DSC)

Figure 3 shows the typical differential scanning calorimetry thermograms of low- and high-sodium pork gels produced without pressure treatment (0.1 MPa) and under high-pressure treatment (150–200 MPa). Three major endothermal transitions were observed that are generally associated with myosin (51–56 °C), sarcoplasmic and connective tissue proteins (60–63 °C), and actin (71–74 °C) [42]. Three main endothermal transitions were observed in the pork gel without any treatment. The ΔH of the endothermal transition decreased with increasing sodium concentration, which indicated a decrease in the total enthalpy of denaturation of the protein occurred with continuous denaturation. The results were 0.206 J/g for the 0% NaCl + 0% SPP gel treated at 0.1 MPa, 0.071 J/g for the 1% NaCl + 0.5% SPP gel treated at 0.1 MPa, and 0.010 J/g for the 2% NaCl + 0.5% SPP gel treated at 0.1 MPa. The peaks for myosin and actin almost completely disappeared with the addition of 2% NaCl. Pressure treatment of the 1% NaCl + 0.5% SPP pork gel resulted in similar denaturation of the proteins, with the total enthalpy of denaturation decreasing with increases in the pressure as follows: 0.071 J/g at 0.1 MPa, 0.057 J/g at 150 MPa, and 0.039 J/g at 200 MPa. The differential scanning calorimetry data showed that denaturation of myofibrils increased at higher pressure, and this denaturation promoted gelation of myofibrils.

### 3.7. Protein Solubility and SDS-PAGE Analysis

The protein solubilities in the pork gels increased with increases in the sodium concentration and pressure. High-pressure treatment led to the swelling of myofibrils, which ultimately resulted in them breaking into shorter filaments. This release of proteins, alongside pressure-induced unfolding of soluble proteins, may contribute to the enhanced gelation and binding effects observed with high-pressure processing [43]. The protein compositions of the supernatants of the unheated pork gels were analyzed using SDS-PAGE (Figure 4). With added sodium chloride and/or SPP, bands were observed at approximately 160 kDa, which indicated increased solubilization of myofibrillar proteins and a higher soluble protein content. The 160 kDa M-protein, located in the M-line, disappears upon sodium chloride addition, which is closely linked to the breakdown of myofibrillar structure [19,29].

After high-pressure treatment within the range of 100–200 MPa, the SDS-PAGE images showed a gradual reduction in the α-actinin band density with increasing pressure (Figure 4). This pattern is consistent with findings from Iwasaki et al. [19], Villacís et al. [29], and Suzuki et al. [36] and suggests that the insolubility of α-actinin may facilitate heat-induced gelation by weakening the myofibrillar structure. Analysis of pork gel supernatants before heating also showed that extracts at 0.1 MPa and without sodium chloride/SPP contained sarcoplasmic proteins, myosin heavy and light chains, and actin. The extraction of myosin and actin slightly increased with sodium chloride addition, likely because of the combined effects of mechanical action and sodium addition. Applying pressures of 150–200 MPa resulted in a notable decrease in the α-actinin band density, which suggested that insoluble α-actinin may contribute to the formation of a dense gel network by destabilizing the Z-line and dispersing myofilaments. These results are consistent with previous research on chicken myofibrils and pressurized pork gels containing 0.2 M NaCl [19]. The reduced extraction rates of certain proteins post-high-pressure treatment imply protein denaturation, which may promote gelation. These findings indicate that the increased solubility of proteins such as myosin and actin, along with other structural proteins, enhances muscle binding, produces a more cohesive structure, and reduces cooking-induced water loss, which improves water retention.

## 4. Conclusions

Reduced sodium/phosphate levels and high pressure improved breaking stress, elastic modulus, hardness, and cohesiveness. Our results showed that high-pressure treatment within the range of 150–200 MPa greatly improved the breaking stress (increased by 10.01% under 150 MPa and 14.66% under 200 MPa), modulus of elasticity (increased by 14.77% under 150 MPa and 24.17% under 200 MPa), and hardness (increased by 11.12% under 150 MPa and 11.45% under 200 MPa). Rheological measurements revealed that gel strength was highest at 150 MPa (G′ = 443,000 Pa; G″ = 66,300 Pa; tanδ = 0.15), which showed higher G′ and G″ values and similar tanδ compared to the 0.1 MPa, 2% NaCl + 0.5% SPP condition (G′ = 334,000 Pa; G″ = 49,200 Pa; tanδ = 0.148). Scanning electron microscopy confirmed that a dense network formed in 1% NaCl pork gels treated at 150–200 MPa, improving microstructural integrity. Additionally, high pressure combined with sodium addition facilitated muscle fiber dissolution and reduced myosin denaturation enthalpy, supporting observed texture improvements. The application of high-pressure treatment within the range of 150–200 MPa significantly reduced cooking loss while maintaining moisture content and provided an ideal network structure for reduced-sodium pork gels. These findings underscore the potential of high-pressure processing as an effective approach for developing high-quality, reduced-sodium meat products, with improvements highlighting its capability to meet the demands for reduced-sodium formulations without compromising product quality. In future work, we aim to conduct further studies on the use of high-pressure treatment within the range of 150–200 MPa to reduce another additive SPP in this experiment, which would provide constructive research for the production of low-sodium, low-phosphate pork products.

## Figures and Tables

**Figure 1 foods-14-00096-f001:**
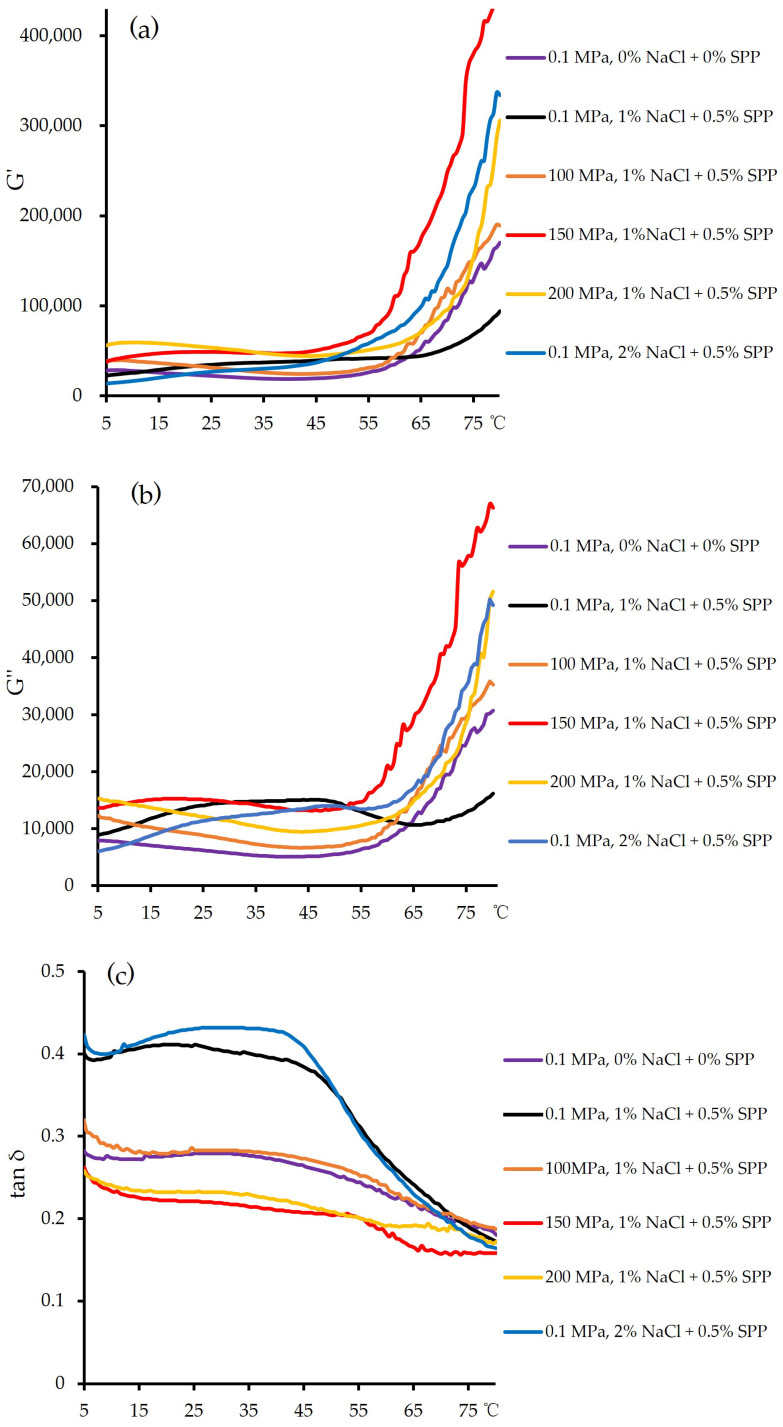
Effect of high-pressure treatment on the G′ (**a**), G″ (**b**), and tanδ (**c**) values of raw pork gels.

**Figure 2 foods-14-00096-f002:**
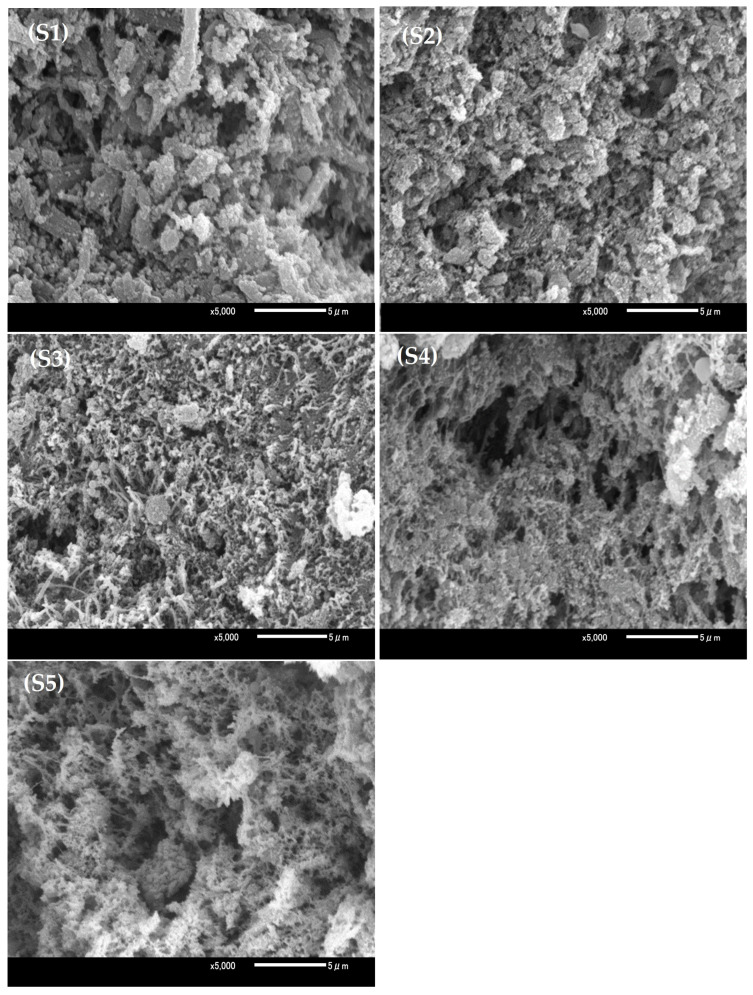
Effect of high-pressure treatment on the microstructure of thermal pork gels. Treatment conditions: (**S1**), 0.1 MPa, 0% NaCl + 0% SPP; (**S2**), (**S3**), and (**S4**): 1% NaCl + 0.5% SPP, and 0.1, 150, or 200 MPa, respectively; and (**S5**): 0.1 MPa, 2% NaCl + 0.5% SPP.

**Figure 3 foods-14-00096-f003:**
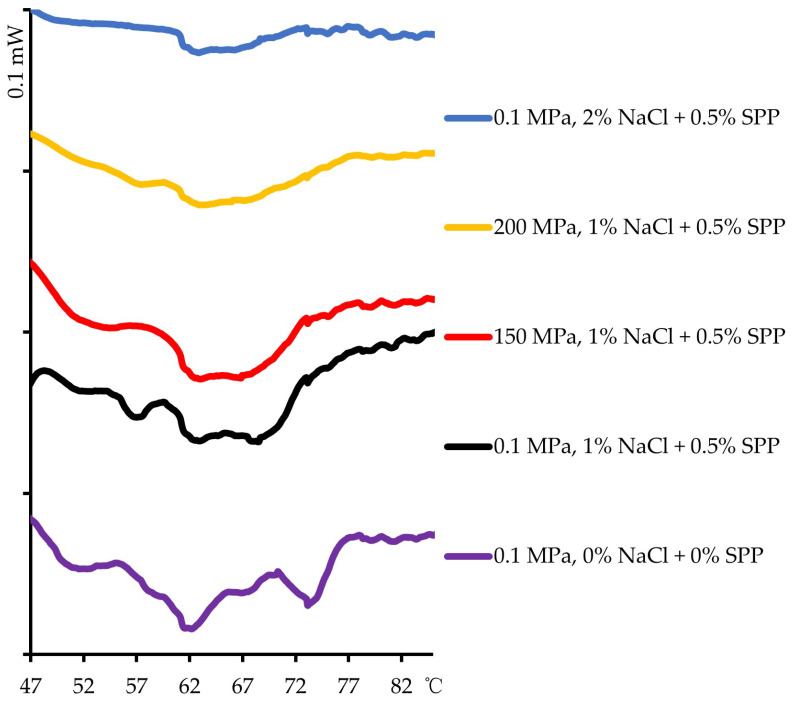
Differential scanning calorimetry thermograms of untreated pork gel, 1% NaCl pork gel treated at 0.1, 150, or 200 MPa, and 2% NaCl pork gel treated at 0.1 MPa. The *y*-axis scale divisions are equivalent to 0.1 mW.

**Figure 4 foods-14-00096-f004:**
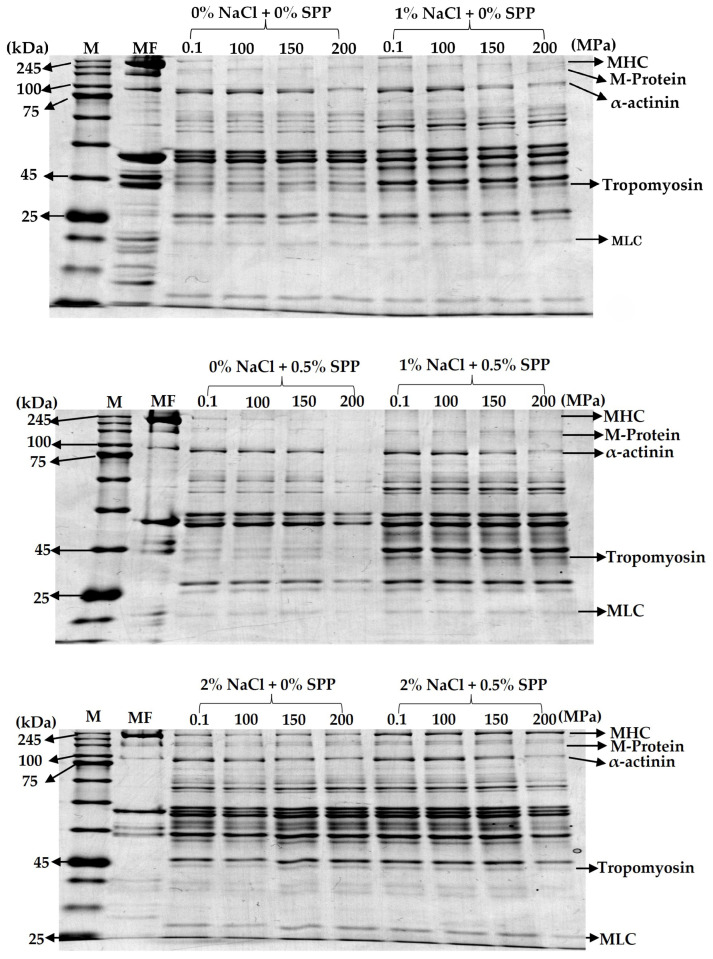
Effect of high-pressure treatment on the SDS-PAGE profiles of supernatants of pork gels. The protein composition of the supernatant fraction was analyzed by electrophoresis on 10% polyacrylamide gel. M: molecular weight marker, MF: myofibrillar proteins, MHC: myosin heavy chain, and MLC: myosin light chain.

**Table 1 foods-14-00096-t001:** Formulation of the pork gels.

Pork Leg (g)	Salt Level (g)	Phosphate Level (g)
60	0.0 (0%)	0.0 (0%)
60	0.0 (0%)	0.3 (0.5%)
60	0.6 (1%)	0.0 (0%)
60	0.6 (1%)	0.3 (0.5%)
60	1.2 (2%)	0.0 (0%)
60	1.2 (2%)	0.3 (0.5%)

Notes: 20 g of pork was added to each metal case and high-pressure treatment was conducted at 0.1, 100, 150, or 200 MPa.

**Table 2 foods-14-00096-t002:** Ingredients of biuret reagent.

Ingredient	Amount
Distilled water	fill up (1 L)
Potassium sodium tartrate	6 g
KI	2 g
${\mathrm{CuSO}}_{4}\cdot 5H_{2}O$	1.5 g
10% NaOH	300 mL

**Table 3 foods-14-00096-t003:** Effect of high-pressure treatment on the color of thermal pork gels.

Parameter	L*	a*	b*
0% NaCl + 0% SPP			
0.1 MPa	$76.48\pm 0.11^{\;a}$	$1.64\pm 0.07^{\;b}$	$10.99\pm 0.06^{\;c}$
100 MPa	$75.09\pm 0.70^{\;b}$	$2.09\pm 0.17^{\;a}$	$11.69\pm 0.08^{\;a}$
150 MPa	$75.15\pm 0.73^{\;b}$	$1.98\pm 0.17^{\;a}$	$11.29\pm 0.14^{\;ab}$
200 MPa	$74.95\pm 0.32^{\;b}$	$2.06\pm 0.09^{\;a}$	$11.47\pm 0.21^{\;ab}$
1% NaCl + 0% SPP			
0.1 MPa	$72.23\pm 1.38^{\;d}$	$2.21\pm 0.36^{\;a}$	$10.82\pm 0.22^{\;cd}$
100 MPa	$73.59\pm 0.52^{\;bc}$	$2.02\pm 0.16^{\;a}$	$10.57\pm 0.04^{\;c}$
150 MPa	$73.24\pm 0.43^{\;d}$	$2.14\pm 0.13^{\;a}$	$10.30\pm 0.03^{\;cd}$
200 MPa	$73.33\pm 1.05^{\;d}$	$2.01\pm 0.11^{\;a}$	$9.87\pm 0.08^{\;c}$
2% NaCl + 0% SPP			
0.1 MPa	$74.76\pm 0.17^{\;c}$	$1.89\pm 0.05^{\;b}$	$10.25\pm 0.10^{\;d}$
100 MPa	$73.00\pm 1.41^{\;d}$	$2.20\pm 0.21^{\;a}$	$10.25\pm 0.03^{\;d}$
150 MPa	$73.41\pm 0.05^{\;cd}$	$2.07\pm 0.06^{\;a}$	$9.97\pm 0.10^{\;d}$
200 MPa	$74.42\pm 0.25^{\;cd}$	$1.88\pm 0.02^{\;b}$	$9.83\pm 0.19^{\;d}$
0% NaCl + 0.5% SPP			
0.1 MPa	$75.87\pm 0.19^{\;abc}$	$0.93\pm 0.04^{\;cd}$	$12.30\pm 0.10^{\;a}$
100 MPa	$73.75\pm 0.67^{\;bc}$	$1.24\pm 0.12^{\;bc}$	$11.12\pm 0.33^{\;bc}$
150 MPa	$76.14\pm 0.39^{\;a}$	$0.78\pm 0.049^{\;d}$	$11.15\pm 0.36^{\;b}$
200 MPa	$75.56\pm 0.09^{\;a}$	$1.27\pm 0.01^{\;bc}$	$11.63\pm 0.11^{\;ab}$
1% NaCl + 0.5% SPP			
0.1 MPa	$75.98\pm 0.09^{\;ab}$	$0.75\pm 0.01^{\;d}$	$11.04\pm 0.05^{\;b}$
100 MPa	$75.05\pm 0.15^{\;b}$	$0.94\pm 0.01^{\;c}$	$10.26\pm 0.08^{\;c}$
150 MPa	$75.04\pm 0.08^{\;b}$	$1.09\pm 0.01^{\;c}$	$10.39\pm 0.09^{\;c}$
200 MPa	$75.38\pm 0.38^{\;b}$	$1.02\pm 0.02^{\;c}$	$10.11\pm 0.04^{\;d}$
2% NaCl + 0.5% SPP			
0.1 MPa	$74.91\pm 0.04^{\;ab}$	$1.08\pm 0.04^{\;c}$	$10.65\pm 0.04^{\;d}$
100 MPa	$73.38\pm 0.27^{\;c}$	$1.19\pm 0.17^{\;bc}$	$11.26\pm 0.14^{\;bc}$
150 MPa	$73.87\pm 0.20^{\;c}$	$1.28\pm 0.24^{\;b}$	$11.36\pm 0.16^{\;b}$
200 MPa	$73.19\pm 0.89^{\;d}$	$1.49\pm 0.08^{\;b}$	$10.65\pm 0.19^{\;d}$

*Note:* All measurements are expressed as the mean ± standard deviation (*n* = 9). ^*a–d*^ Different parameter superscripts indicate significant differences (p<0.05).

**Table 4 foods-14-00096-t004:** Effect of high-pressure treatment on the water content and cooking losses of thermal pork gels.

Parameter	Water Content (%)	Cooking Losses (%)
0% NaCl + 0% SPP		
0.1 MPa	$68.97\pm 1.30^{\;a}$	$18.83\pm 0.07^{\;a}$
100 MPa	$71.44\pm 2.56^{\;a}$	$19.67\pm 0.17^{\;a}$
150 MPa	$70.48\pm 2.53^{\;a}$	$19.89\pm 0.17^{\;a}$
200 MPa	$71.08\pm 3.02^{\;a}$	$18.77\pm 0.09^{\;a}$
1% NaCl + 0% SPP		
0.1 MPa	$68.59\pm 0.82^{\;a}$	$13.97\pm 0.36^{\;ab}$
100 MPa	$70.58\pm 1.78^{\;a}$	$13.49\pm 0.16^{\;b}$
150 MPa	$69.82\pm 1.09^{\;a}$	$15.37\pm 0.13^{\;b}$
200 MPa	$70.53\pm 0.70^{\;a}$	$15.19\pm 0.11^{\;b}$
2% NaCl + 0% SPP		
0.1 MPa	$67.96\pm 1.72^{\;a}$	$11.98\pm 0.05^{\;bc}$
100 MPa	$69.12\pm 2.24^{\;a}$	$13.31\pm 0.21^{\;b}$
150 MPa	$69.69\pm 2.05^{\;a}$	$12.36\pm 0.06^{\;b}$
200 MPa	$69.65\pm 2.25^{\;a}$	$13.25\pm 0.02^{\;b}$
0% NaCl + 0.5% SPP		
0.1 MPa	$69.46\pm 2.08^{\;a}$	$15.19\pm 0.04^{\;b}$
100 MPa	$71.03\pm 1.57^{\;a}$	$19.02\pm 0.12^{\;a}$
150 MPa	$70.30\pm 1.37^{\;a}$	$19.15\pm 0.049^{\;a}$
200 MPa	$70.68\pm 0.72^{\;a}$	$17.94\pm 0.01^{\;ab}$
1% NaCl + 0.5% SPP		
0.1 MPa	$69.94\pm 1.51^{\;a}$	$9.79\pm 0.01^{\;c}$
100 MPa	$71.28\pm 0.99^{\;a}$	$8.68\pm 0.01^{\;cd}$
150 MPa	$70.85\pm 0.43^{\;a}$	$8.64\pm 0.01^{\;cd}$
200 MPa	$71.32\pm 1.06^{\;a}$	$8.95\pm 0.02^{\;c}$
2% NaCl + 0.5% SPP		
0.1 MPa	$70.92\pm 2.40^{\;a}$	$7.52\pm 0.04^{\;d}$
100 MPa	$71.00\pm 2.74^{\;a}$	$6.98\pm 0.17^{\;de}$
150 MPa	$70.48\pm 2.93^{\;a}$	$6.64\pm 0.24^{\;e}$
200 MPa	$70.46\pm 2.45^{\;a}$	$7.11\pm 0.08^{\;d}$

*Note*: All measurements are expressed as the mean ± standard deviation (*n* = 9). ^*a–e*^ Different parameter superscripts indicate significant differences (p<0.05).

**Table 5 foods-14-00096-t005:** Effect of high-pressure treatment and NaCl/SPP addition on the breaking stress, elastic modulus, hardness, and cohesiveness of thermal pork gels.

Parameter	Breaking Stress ($\times 10^{5}$ Pa)	Modulus of Elasticity ($\times 10^{6}$ Pa)	Hardness ($\times 10^{5}$ N)	Cohesiveness
0% NaCl + 0% SPP				
0.1 MPa	$4.46\pm 0.49^{\;a}$	$8.00\pm 0.25^{\;b}$	$3.11\pm 0.03^{\;ab}$	$0.39\pm 0.03^{\;d}$
100 MPa	$4.38\pm 0.29^{\;a}$	$8.95\pm 1.87^{\;ab}$	$3.11\pm 0.43^{\;ab}$	$0.40\pm 0.03^{\;cd}$
150 MPa	$5.15\pm 0.45^{\;a}$	$10.73\pm 0.74^{\;ab}$	$3.23\pm 0.06^{\;ab}$	$0.42\pm 0.03^{\;c}$
200 MPa	$4.60\pm 0.50^{\;a}$	$11.23\pm 1.36^{\;a}$	$3.43\pm 0.30^{\;ab}$	$0.42\pm 0.01^{\;d}$
1% NaCl + 0% SPP				
0.1 MPa	$2.51\pm 0.41^{\;d}$	$6.12\pm 0.65^{\;d}$	$2.43\pm 0.13^{\;c}$	$0.44\pm 0.01^{\;c}$
100 MPa	$2.73\pm 0.58^{\;d}$	$6.98\pm 1.34^{\;c}$	$2.25\pm 0.26^{\;d}$	$0.42\pm 0.01^{\;d}$
150 MPa	$2.97\pm 0.38^{\;bc}$	$7.17\pm 2.08^{\;bc}$	$2.49\pm 0.32^{\;c}$	$0.43\pm 0.01^{\;c}$
200 MPa	$3.62\pm 0.10^{\;b}$	$9.23\pm 0.58^{\;ab}$	$3.21\pm 0.48^{\;ab}$	$0.50\pm 0.02^{\;a}$
2% NaCl + 0% SPP				
0.1 MPa	$3.08\pm 0.05^{\;bc}$	$6.39\pm 0.76^{\;d}$	$2.91\pm 0.27^{\;bc}$	$0.44\pm 0.03^{\;c}$
100 MPa	$2.57\pm 0.34^{\;d}$	$6.60\pm 0.12^{\;cd}$	$3.47\pm 0.23^{\;ab}$	$0.44\pm 0.02^{\;c}$
150 MPa	$3.18\pm 0.40^{\;b}$	$8.55\pm 0.70^{\;ab}$	$3.25\pm 0.33^{\;ab}$	$0.43\pm 0.03^{\;c}$
200 MPa	$3.72\pm 0.13^{\;ab}$	$10.93\pm 0.70^{\;ab}$	$2.73\pm 0.21^{\;bc}$	$0.43\pm 0.02^{\;cd}$
0% NaCl + 0.5% SPP				
0.1 MPa	$4.50\pm 0.27^{\;a}$	$7.41\pm 3.22^{\;bc}$	$3.21\pm 0.03^{\;ab}$	$0.49\pm 0.01^{\;ab}$
100 MPa	$4.77\pm 0.61^{\;a}$	$7.21\pm 1.84^{\;bc}$	$3.94\pm 0.14^{\;a}$	$0.48\pm 0.02^{\;ab}$
150 MPa	$4.14\pm 0.48^{\;b}$	$8.22\pm 2.37^{\;b}$	$3.73\pm 0.14^{\;ab}$	$0.46\pm 0.03^{\;b}$
200 MPa	$4.76\pm 0.46^{\;a}$	$8.34\pm 1.55^{\;b}$	$3.51\pm 0.01^{\;ab}$	$0.45\pm 0.01^{\;bc}$
1% NaCl + 0.5% SPP				
0.1 MPa	$2.78\pm 0.16^{\;cd}$	$8.11\pm 0.31^{\;b}$	$2.91\pm 0.39^{\;b}$	$0.43\pm 0.02^{\;cd}$
100 MPa	$3.38\pm 0.18^{\;c}$	$7.68\pm 1.27^{\;b}$	$3.44\pm 0.45^{\;ab}$	$0.44\pm 0.01^{\;bc}$
150 MPa	$3.14\pm 0.48^{\;c}$	$7.47\pm 1.58^{\;b}$	$3.95\pm 0.68^{\;a}$	$0.47\pm 0.01^{\;b}$
200 MPa	$3.33\pm 0.54^{\;c}$	$8.75\pm 1.37^{\;b}$	$3.52\pm 0.92^{\;ab}$	$0.46\pm 0.01^{\;b}$
2% NaCl + 0.5% SPP				
0.1 MPa	$2.70\pm 0.26^{\;d}$	$7.58\pm 1.73^{\;b}$	$2.91\pm 0.09^{\;b}$	$0.54\pm 0.01^{\;a}$
100 MPa	$2.77\pm 0.46^{\;cd}$	$8.42\pm 1.25^{\;b}$	$2.77\pm 0.13^{\;b}$	$0.52\pm 0.02^{\;a}$
150 MPa	$3.28\pm 0.16^{\;c}$	$8.86\pm 0.80^{\;ab}$	$3.23\pm 0.09^{\;ab}$	$0.53\pm 0.03^{\;a}$
200 MPa	$3.18\pm 0.30^{\;b}$	$9.79\pm 1.51^{\;ab}$	$3.46\pm 0.81^{\;ab}$	$0.50\pm 0.01^{\;a}$

*Note*: All measurements are expressed as the mean ± standard deviation (*n* = 15). ^*a–d*^ Different parameter superscripts indicate significant differences (p<0.05).

## Data Availability

The original contributions presented in this study are included in the article. Further inquiries can be directed to the corresponding author.

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
