# Peer review of "High-Pressure Treatment in Combination with Reduced Sodium for Improving the Physicochemical Properties and Sensory Qualities of Pork Gels"

_foods, 2025, doi:10.3390/foods14010096_

Round 1

Reviewer 1 Report

Comments and Suggestions for Authors

The effects of high-pressure treatment on texture, microstructure, and sensory properties of pork gels with sodium chloride and/or sodium polyphosphate were studied. The writing of the paper is standardized, the data is well detailed, and the research content is complete. However, certain problems need to be carefully revised before the article is accepted.

1. There are some writing problems in this article, please revise accordingly.

(1)The relative line spacing of 200 MPa in Table 2 is too small, which makes it easy to misunderstand the letters when they are blocked;

(2)Some years in the references are not bolded, please check the uniform format of all the references.

2. The water retention data in Table 3 show no significant difference in pressure and sodium concentration. How can we conclude that the water retention is improved? Can we discuss the reasons why the addition of sodium chloride or phosphate in 3.2 enhances the water retention capacity of the pork gel treated under high pressure?

3. As shown in Table 4, the effect of sodium concentration on cohesion is quite different. Can we discuss the effect of sodium concentration on the cohesion of pork gel?

4. Two variables were used in the experiment in terms of additives, and differences in sodium concentration were also analyzed. However, the paper mainly focused on high-pressure treatment, and could it increase the analysis of the influence of SPP on pork gel? In this way, I believe it will help to further enhance the innovation and depth of this paper.

5. S1 in SEM is not marked in the picture, so Figure 3 can be reformatted to improve readability. Other pictures can also be reformatted as appropriate.

6. The content of the mechanism discussed in this paper is not deep enough. A comparison of the results of the physicochemical properties of pork gels may not be enough.

7. The following literature shows that his research is relatively rich and can be properly referred to:

Transglutaminase cross-linking ovalbumin-flaxseed oil emulsion gels: Properties, microstructure, and performance in oxidative stability [J]. Food Chemistry, 2024, 448: 138988.

Reviewer 2 Report

Comments and Suggestions for Authors

Many statements were made without references or additional information for substantiation of their veracity. There are many missing details that prevent determination if the materials and methods were appropriate and do not allow other scientists to duplicate the experimental conditions.

Line(s)      Comment

1-4            Abstracts usually only need a one sentence introduction to the research.

7-14          Inclusion of data would substantiate the results.

20-24        Reference(s) needed to substantiate that this information is factual.

28-30        Reference(s) needed to substantiate that this information is factual.

34-40        Reference(s) needed to substantiate that this information is factual.

43-48        Run-on sentence.

50-52        This sentence is misplaced; move to l. 75.

56-60        Reference(s) needed to substantiate that this information is factual.

75-84        This information does not add to the introduction of the research and should be moved to the discussion section when results of the current study are compared with those of other studies.

94              The muscles from the pork leg should be described.

96              The method of removing the connective tissue and fat and the size of the pieces must be given.

97              The level of vacuum, model and manufacturer of the vacuum sealer, thickness or vapor permeability of the polyethylene bags, and source of the polyethylene bags must be given.

99-100      The type of probe and model and manufacturer of the pH meter must be given.

105           The source of the sodium polyphosphate must be given.

114           The thickness or vapor permeability and source of the polyethylene bags must be given.

126           The aperture opening size, optical geometry, observer angle, and illuminant type must be given.

133-134   The equation or a reference for the equation must be given.

134, 138, 146 The description of the 24 groups must be given, e.g. 3 pork legs  x 2 gel types (raw and cooked) x 6 treatments (salt, SPP) = 36 groups.

174           The scientific sensory guidelines ASTM Food and Beverage Sensory Evaluation Standards, American Meat Science Association Research Guidelines for Cookery, Sensory Evaluation, and Instrumental Tenderness Measurements of Meat, IFST Guidelines for Ethical and Professional Practices for the Sensory Analysis of Foods, IFT Sensory Evaluation Guide for Testing Food and Beverage Products, ISO sensory analysis standards, Society of Sensory Professionals Recommendations for Publications Containing Sensory Data should be reviewed as 10 members are not acceptable for an untrained panel.

181           The conditions for testing, including the number of sessions for the assessment of two different gels; environment of separate booths, lighting, temperature; method of coding the samples, and other parameters, must be given.

205           A reference or more detailed description of the Biuret determination must be given.

224-234   It should be explained how this procedure is different than the procedure described in l. 212-223.

236-239   Additional details of the statistical analyses are needed, including the experimental design, model for the ANOVA, and specific Tukey tests.

241           This section is “Results and Discussion” since discussion of the results is in each subsection.

252-254   The method of determining the smoothness and hydration of the heat treated samples must be described in the materials and methods section.

Tables 2, 3, 4 If the means were appropriately separated by Tukey procedures, then a single set of letters (capitalized or uncapitalized) could be used to indicate the differences among the 24 means for each measurement. Tukey’s test compares the mean of each treatment to the mean of every other treatment to identify differences between the two means that is significantly larger than the expected standard error.

Table 2 footnote Use of “significantly” and the probability level in the same sentence in the footnote and in subsequent sentences is redundant.

295-296   This assumption is valid only if substantiated by reference(s) or the photomicrographs, of which the results have not yet been presented in the paper.

326-328   Reference(s) needed to substantiate that high pressure would increase exposure of the groups and enhance crosslinking during heating.

343-344   Reference(s) needed to substantiate that this information is factual.

346-352   Reference(s) needed to substantiate that this information is factual.

366           The determination of acceptability is valid only if more than 50 untrained members evaluate the product.

376-377   Reference(s) needed to substantiate that this information is factual.

445           This section should be “Conclusions.”

445-463   A conclusions section has only a few of the most important results and relates those to how the results can be used, uniqueness of the results, and what other studies should be done based on the results.

501           International Journal of Epidemiology”

Round 2

Reviewer 2 Report

Comments and Suggestions for Authors

Journal articles to convey scientific research results require precise language and specific explanations, both of which are missing in many instances in the revised manuscript. The information must be as complete and useful as possible to a scientist and/or processor.

Line(s)      Comment

1                Delete one of the “in this study” phrases.

4                Delete “We investigated” to provide proper sentence grammar.

6-7            Incomplete sentence.

9                It should be clarified that the high-pressure treatment caused the higher values compared with the control.

16              The word “healthier” should be deleted because the health of humans was not measured in this study.

17-18        “good” is not descriptive and consumer expectations and dietary guidelines were not measured in this study.

27-28        Reference(s) needed to substantiate that there has been a notable decline in meat and meat product consumption over the past two decades because the three references discuss meat in the diet and not consumption patterns of meat.

32-33        Reword; “Between 1982 and 2015, urine samples indicated the average sodium chloride (NaCl) intake at 9.3 g/day [8].”

33-34        Reference(s) needed to substantiate the WHO recommendation on NaCl intake per day.

36-40        Reference(s) needed to substantiate that this information is factual.

37              “includes items”

86              “inside muscle of three pork legs, which were processed“; the specific muscle name should be given.

128           “across 24 treatment groups (6 formulations x 4 high-pressure treatments), with three replicates per group.”

228-231    This sentence is unacceptable unless the methods of determining smoothness and hydration are described in the materials and methods section or if the results of the SEM photomicrographs have been previously presented.

234           “Table 3”

Tables 3, 4, 5 Tukey procedures compare the means of each treatment with the mean of every other treatment so a single set of letters should be used to indicate differences among the means for each variable. This provides the necessary information for a processor or scientist to compare means of two different combinations of high-pressure, NaCl, and SPP is desired for a specific trait, e.g. 100 MPa/0% NaCl/0.5% SPP to 150 MPa/2% NaCl/0.5% SPP.

248           “Table 4”

261-262    Comparing each mean with the other 23 means would provide the statistical probability of difference among these treatments.

268           “Table 5”

272-273    This attribution cannot be given because the results of the microstructure determination have not yet been presented in the manuscript.

330-336    Reference(s) needed to substantiate that this information is factual.

341           “Figure 2”

363           Correlation in scientific terms implies a statistical analysis to determine the probability of the relationship between the variables so the correlation analysis should be described in the statistical analysis section of the materials and methods and the specific correlation data should be given here.

368           “Figure 3”

392, 398   “Figure 4”

417-421   The conclusion summarizes the results without giving specific data.

428           Sensory characteristics were not reported in the revised manuscript.

429           It was not established in the manuscript that lower sodium meat products are healthier for humans.
